# Learning to Allocate Propagating Treatments: Maximize Uplift under Network Interference

## Abstract

Uplift modeling and treatment allocation are classical tasks in promotion marketing. Yet existing allocations ignore propagating treatments and network interference, where both outcomes and the propagation mechanism vary with peers' treatment history, making policy value hard to estimate and optimize. We formalize a history-driven uplift objective with activation probability $g(\boldsymbol{Z}_i^t, \boldsymbol{X}_i)$ and outcomes that depend on neighbors' treated states. Theoretically, we establish conditions for identification and provide finite-sample guarantees for policy evaluation under interference and model misspecification. Methodologically, we propose GUM-DT via a Monte-Carlo policy search: learn an ensemble of lightweight propagation models and an outcome model, and evaluate candidate allocations via double-robust (DR) estimators with IPW corrections. On synthetic networks, experiments demonstrate consistent gains over uplift allocations of GUM-DT, validating robustness and effectiveness.

## 1 Introduction

Treatments in modern marketplaces often *propagate* over social graphs—referral-driven memberships, friend-invite trials, community bundles—where both activation and revenue *evolve with* neighbors' treatment states. We study how to allocate an initial budget of seeds on a network $G = (V, E)$ to maximize *network-wide uplift* relative to a non-zero baseline policy $\pi_0$, where activations and outcomes unfold over $T$ periods under a history-dependent propagation mechanism $g$ and outcome mechanism $m$. This objective departs from independent-unit or non-propagating settings by *jointly* modeling activation dynamics and the payoffs they induce through neighbors' evolving states. A representative case is a paid-membership program with peer referrals: "join-membership" can spread over time, and a user's incremental revenue depends on the contemporaneous and cumulative membership of friends—exhibiting complementarities (co-purchases, retention) or crowding/fatigue.

To position our problem, we contrast it with three nearby paradigms. Individual Treatment Regime (*ITR*) selects $x \mapsto a$ per individual while abstracting from network exposure (Zhao et al., 2012; Athey & Wager, 2021). *Static uplift* ranks one-shot ROI under a budget, assuming treatments do not spread (*non-propagating*). *Influence maximization (IM)* optimizes expected spread rather than causal value (*proxy misalignment*) (Kempe et al., 2003). In contrast, we optimize uplift relative to a non-zero baseline when both activation and payoffs are history-dependent, which breaks submodularity and invalidates classical spread-based greedy guarantees.

Our setting relates to *approximate neighborhood interference* (ANI)—exposure-mapping approaches under local interference (Leung, 2022). Both frameworks formalize how neighbors' treatment states shape outcomes. The distinction is twofold. First, ANI focuses on *estimation* of exposure-defined effects at a given time, whereas we address *propagating* treatments whose activation trajectories are *endogenous to the seed set* and target *policy optimization* for uplift over a non-zero baseline. Second, ANI exposures are typically static or period-wise, while our objective *aggregates over history*, inducing non-submodularity and non-stationary marginal gains for which one-shot ranking and classical greedy are inadequate. In short, we leverage ANI-style locality to

parameterize and learn exposure-sensitive mechanisms, but depart in targeting network-wide uplift under propagation and in coupling estimation with an optimizer aligned to history dependence.

Two challenges arise naturally. On the estimation side, $\text{Value}(\pi)$ integrates over an exponential family of latent activation paths; $g$ is unknown and history dependent; outcomes may be non-monotone in neighborhood treatment. On the optimization side, each seed perturbs the environment seen by the others, so marginal gains are non-stationary; heuristics that assume fixed gains or rely on spread proxies lack guarantees. We address these issues with a two-layer framework. An evaluation layer learns an ensemble of lightweight propagation models $\{\hat{g}\}$ and an outcome model $\hat{m}$, and scores candidate allocations using a targeted doubly robust (DR) estimator that combines inverse propensity weighting with TMLE-style targeting; the estimator is consistent if either $\hat{g}$ or $\hat{m}$ is correctly specified and attains improved finite-sample stability via ensembling and targeting. An optimization layer, GUM-DT, iteratively queries this targeted DR oracle under the current environment and refreshes stale gains before selection, aligning the search procedure with the history-dependent objective rather than a spread proxy.

Our contributions are threefold. (i) *Formulation.* We formalize causal uplift maximization with propagating treatments on networks, defining value as uplift relative to a non-zero baseline under history-dependent interference and clarifying its relation to, and difference from, ANI-style exposure models. (ii) *Robust off-policy evaluation.* We develop a targeted DR estimator tailored to propagation settings that is doubly robust and more stable in finite samples through ensembling and targeting. (iii) *History-aligned optimization and theory.* We design GUM-DT, a dynamic-greedy allocator that refreshes marginal gains via the targeted DR oracle, and provide PAC-style analyses that decompose policy regret into model misspecification, statistical estimation, and algorithmic approximation errors, accompanied by diagnostics for exposure non-monotonicity and misspecified propagation. Together, these elements provide a principled bridge from causal objectives to practical, high-stakes allocation of propagating treatments in networked markets.

## 2 RELATED WORKS

Our research synthesizes three distinct but complementary lines of work: causal effect estimation on networks, off-policy policy learning, and Influence Maximization (IM). Details of off-policy policy learning and IM part are in Appendix. A significant body of works address treatment effect estimation under network interference, where SUTVA is violated and treatment effects depend on peers' treatment and exposure (Hudgens & Halloran, 2008; Aronow & Samii, 2017). Recent methods model exposure mappings and learn representations of neighborhood history to estimate direct and spillover effects (Ma & Tresp, 2021; Guo et al., 2022). These estimators can be adapted into uplift–first baselines (e.g., estimate CATE or ITE (Shalit et al., 2017), then rank under a budget). While effective for estimation, they do not by themselves address the *policy design* problem with propagating actions: evaluating and optimizing an initial seed set whose value depends on the distribution of activation trajectories and on path–dependent outcomes.

## 3 PROBLEM FORMULATION

We first define basic notations and objective, then detail the system dynamics that govern the intervention process and state the assumptions for identification.

### 3.1 NOTAIONS AND OBJECTIVE

Let $G(V, E)$ be a graph observed over a discrete time horizon $t = 0, 1, \ldots, T$. An intervention seeding policy, $\pi_S$, selects an initial seed set $S \subseteq V$ of size at most $K$. This policy initiates a stochastic propagation process, resulting in a full activation path over the network, denoted $\overline{\boldsymbol{A}}_{\pi_S} = (\boldsymbol{A}^0, \ldots, \boldsymbol{A}^T)$, where $\boldsymbol{A}^t := (A_1^t, \ldots, A_{|V|}^t) \in \{0, 1\}^{|V|}$ is the vector of binary activation states of all nodes at time $t$. We write $\mathbf{H}^t = (\boldsymbol{A}^0, \ldots, \boldsymbol{A}^t)$ for the global history up to $t$, and $\mathbf{H}_U^t$ for its restriction to nodes $U \subseteq V$. Features of nodes are denoted as $\boldsymbol{X} = \{\boldsymbol{X}_i : i \in V\}$.

**Definition 1** (Path-dependent Potential Outcome). *For any realized activation path $\overline{\boldsymbol{A}}$ on graph $G$, the potential outcome of node $i$ is $Y_i(\overline{\boldsymbol{A}}) \in \mathbb{R}$, which may depend on on the entire path (e.g., fatigue or timing effects), not only on terminal activation.*

**Definition 2** (Policy Value: Net Uplift). *Given a seeding policy $\pi_S$ (choosing an initial seed set $S \subseteq V$, $|S| \leq K$) and the baseline policy $\pi_\varnothing$, define*

$$V(\pi_S) := \mathbb{E}_{\overline{\boldsymbol{A}} \sim \mathbb{P}(\cdot \mid \pi_S, \boldsymbol{X})}\Big[ \sum_{i=1}^n Y_i(\overline{\boldsymbol{A}}) \Big] - \mathbb{E}_{\overline{\boldsymbol{A}} \sim \mathbb{P}(\cdot \mid \pi_\varnothing, \boldsymbol{X})}\Big[ \sum_{i=1}^n Y_i(\overline{\boldsymbol{A}}) \Big].$$

## 3.2 System Dynamics and Identification

**Definition 3** (Propagation Mechanism). *For $t \geq 1$, the state of any inactive node $v_i$ evolves according to a conditional probability function $g$ depending on the node's features $\boldsymbol{X}_i$ and an exposure vector $\boldsymbol{Z}_i^t$, which summarizes the activation history of its neighbors $\mathcal{N}(i)$.*

$$\mathbb{P}(A_i^t = 1 | \boldsymbol{H}^{t-1}, \boldsymbol{X}) = g(\boldsymbol{Z}_i^t, \boldsymbol{X}_i), \quad \text{where } \boldsymbol{Z}_i^t = h(\boldsymbol{X}_i, \boldsymbol{H}_{\mathcal{N}(i)}^{t-1}) \tag{1}$$

We assume that an activated node remains active, $A_i^{t-1} = 1 \Rightarrow A_i^t = 1$. This allows history–dependent, possibly non–monotone peer effects, where, for instance, excessive neighbor activity could decrease activation probability. Classic models like Independent Cascade (IC) or Linear Threshold (LT) arise as special cases.

**Definition 4** (Outcome Mechanism). *The expected total utility is determined by node features and nodes treated history*

$$\mathbb{E}\left[ \sum_{i \in V} Y_i | \overline{\boldsymbol{A}}, \boldsymbol{X} \right] = m(\overline{\boldsymbol{A}}, \boldsymbol{X}) \tag{2}$$

where $m$ may be instantiated by a model that captures graph structure and temporal dependence. Intuitively, $g$ induces a distribution over paths, while $m$ assigns value to each realized path.

To connect our causal objective to observational data, we rely on the following standard assumptions, adapted to our dynamic, networked setting.

**Assumption 1** (Consistency). *For the realized path $\overline{\boldsymbol{A}}$ under any intervention seeding, the observed outcome for each node corresponds to the potential outcome under the specific realized activation path that occurred: $Y_i = Y_i(\overline{\boldsymbol{A}})$.*

**Assumption 2** (Positivity). *For any local history that occurs with positive probability, the conditional probability of a node's activation is bounded away from 0 and 1.*

**Assumption 3** (Sequential Ignorability). *Conditional on the observed local history of a node and its neighbors up to time $t-1$, its activation at time $t$ is independent of the set of all potential outcome functions, $\{\mathbf{Y}(\cdot)\}$. Formally:*

$$A_i^t \perp\!\!\!\perp \{\mathbf{Y}(\cdot)\} \mid \boldsymbol{H}_{\{i\} \cup \mathcal{N}(i)}^{t-1}, \boldsymbol{X} \tag{3}$$

*This is the crucial no-unmeasured-confounders assumption, allowing us to treat the sequential propagation as a series of conditionally randomized experiments.*

Together, these assumptions ensure that our theoretical objective is empirically grounded.

**Proposition 1** (Identifiability of Policy Value). *Under Assumptions 1-3, $V(\pi_S)$ is identifiable from observational data.*

*Proof sketch.* By the g-computation formula (Robins, 1986),

$$\mathbb{E}\Big[ \sum_i Y_i(\overline{\boldsymbol{A}}) \Big| \pi_S, \boldsymbol{X} \Big] = \sum_{\boldsymbol{a}} \Big( \sum_i \mathbb{E}[Y_i \mid \overline{\boldsymbol{A}} = \boldsymbol{a}, \boldsymbol{X}] \Big) \mathbb{P}(\overline{\boldsymbol{A}} = \boldsymbol{a} \mid \pi_S, \boldsymbol{X}).$$

By Consistency, the counterfactual expectation equals the observable conditional expectation. By chain rule and Sequential Ignorability given local histories, each factor $\mathbb{P}(A_i^t \mid \boldsymbol{H}_{\{i\} \cup \mathcal{N}(i)}^{t-1}, \boldsymbol{X})$ is identifiable; Positivity ensures they are well-defined. Thus both terms in Def. 2 are identifiable, hence $V(\pi_S)$ is identifiable. $\square$

**Computational Implication.** Although identifiable, direct evaluation of $V(\pi_S)$ is intractable due to the exponential number of activation paths. This motivates estimating $V(\pi_S)$ from data and optimizing allocations using this estimate, while controlling the resulting estimation and approximation errors.

# 4 METHODOLOGY AND THEORETICAL GUARANTEES

Having established that our causal objective is identifiable but intractable to compute, our methodology directly confronts this challenge with a two-stage framework: **estimation** and **optimization**. We first develop robust methods to estimate the policy value from observational data, and then design efficient algorithms to find the optimal seed set based on this estimate.

## 4.1 ESTIMATING POLICY VALUE FROM OBSERVATIONAL DATA

Estimating the policy value $V(\pi_S)$ from observational data situates our problem within the framework of **off-policy evaluation**. We must use data generated under a historical *behavior policy* $\pi_b$ to evaluate our new *target policy* $\pi_S$. This requires learning two key components of the system's dynamics from the available data.

### 4.1.1 LEARNING THE SYSTEM DYNAMICS

**Modeling Choices.** Beyond the core identification assumptions that enable causal reasoning, our estimation framework relies on two modeling choices to build concrete estimators and quantify uncertainty. (M1) We assume that the point estimate for the activation probability $p = \widehat{g}(\boldsymbol{Z}_i^t, \boldsymbol{X}_i)$ follows a Beta distribution. (M2) We assume that the total observed outcome for a given path, $\sum_i Y_i$, follows a Gaussian distribution. These distributional assumptions are choices for the modeling architecture and are not required for causal identification itself.

**Propagation Model.** The propagation probability $g(\boldsymbol{Z}_i^t, \boldsymbol{X}_i)$ is dynamic, depending on the evolving history of network activations. We learn this function by training an ensemble of $M$ lightweight models $\{\widehat{g}^{(m)}\}_{m=1}^M$. The history is encoded into a vector $\boldsymbol{Z}_i^t$ using a Time-Channel GNN, which captures spatio-temporal dependencies. Training an ensemble on bootstrap samples enhances robustness and quantifies model uncertainty. For estimators requiring importance weighting, we similarly train a model of the behavior policy's propagation dynamics, denoted $\widehat{g}_b$. The detailed training procedure is presented in Algorithm 1.

**Outcome Model.** The second component is the outcome model, $\widehat{m}$, which predicts the total network utility given a full activation history, $\mathbb{E}[\sum_i Y_i | \overline{\boldsymbol{A}}, \boldsymbol{X}]$. We implement $\widehat{m}$ as a Graph Neural Network (GNN) to effectively capture how path-dependent dynamics on the underlying graph topology influence the final outcome. This frames the learning problem as a graph-level regression task, mapping the entire history to a single utility score.

### 4.1.2 ESTIMATORS FOR POLICY VALUE

Based on the learned models, we can construct several estimators for $V(\pi_S)$.

**Outcome Regression (OR) Estimator.** The OR estimator (Algorithm 4) is a direct, simulation-based approach. To account for model uncertainty, each simulation rollout uses a propagation model randomly sampled from the trained ensemble. Its consistency, however, stringently requires that *both* the propagation models and the outcome model are correctly specified.

**Inverse Propensity Weighting (IPW) Estimator.** The IPW estimator (Algorithm 5) re-weights observed historical outcomes. To compute the importance weights, it requires a single propagation model for the target policy. We use the average of the ensemble, $\bar{g} = \frac{1}{M} \sum_m \widehat{g}^{(m)}$. IPW's consistency depends only on the correct specification of the propagation models ($\bar{g}, \widehat{g}_b$), but it can suffer from high variance if the target policy evaluates trajectories that were rare under the behavior policy.

**Doubly Robust (DR) Estimator.** The DR estimator (Algorithm 2) synthesizes the OR and IPW approaches to achieve superior statistical properties. It combines a direct simulation-based estimate with an importance-weighted correction term based on the observed outcome residual. We adopt a Monte-Carlo version of the DR estimator for evaluation.

---

**Algorithm 1** PROPAGATIONMODELLEARNING (Training an ensemble of $M$ propagation model $g$)

---

**Require:** Cascades $\{A^{b,0:T_b}\}_{b=1}^{B}$; node features $\boldsymbol{X} = \{\boldsymbol{X}_i\}_{i \in V}$; history encoder $h_\phi$; model family $\mathcal{G}$; ensemble size $M$; optional subsampling rates: time-step rate $\rho_t \in (0, 1]$, negative-node rate $\rho_n \in (0, 1]$.

**Ensure:** Trained ensemble $\{\widehat{g}^{(m)}\}_{m=1}^{M}$.

1: **History encoder $h_\phi$ (Time-Channel GNN).** For each node $i$ and step $t$, compute the history code $\boldsymbol{Z}_i^t$ via:

$$\boldsymbol{E}_i^{t-1} = [\boldsymbol{X}_i, \boldsymbol{H}^{t-1}], \quad \boldsymbol{Z}_i^t = \mathrm{MLP}(\sigma(W_1 \boldsymbol{E}_i^{t-1} + U_1 \sum_{j \in \mathcal{N}(i)} \boldsymbol{E}_j^{t-1} + b_1)).$$

2: **Build supervised dataset $\mathcal{D}$.**
3: $\mathcal{D} \leftarrow \varnothing$
4: **for** $b = 1$ **to** $B$ **do**                                         ▷ iterate cascades
5:     **for** $t = 1$ **to** $T_b$ **do**
6:         **if** $\mathrm{Unif}(0,1) > \rho_t$ **then continue**                 ▷ optional time-step subsampling
7:         **end if**
8:         $\mathcal{I}^{b,t} \leftarrow \{i \in V : A_i^{b,t-1} = 0\}$                     ▷ inactive at $t-1$
9:         **if** negative subsampling enabled **then**
10:            $\mathcal{I}^{b,t} \leftarrow \{i \in \mathcal{I}^{b,t} : A_i^{b,t} = 1\} \cup \{i \in \mathcal{I}^{b,t} : A_i^{b,t} = 0 \wedge \mathrm{Unif}(0,1) \leq \rho_n\}$
11:         **end if**
12:         **for** each $i \in \mathcal{I}^{b,t}$ **do**
13:            $Z_i^t \leftarrow h_\phi(\mathbf{X}, \boldsymbol{H}_{\mathcal{N}(i)}^{t-1,b})$          ▷ encode *local* neighbor history via $h\phi$ in line 1
14:            $y_i^t \leftarrow A_i^{b,t}$                          ▷ label: activated at step $t$?
15:            $\mathcal{D} \leftarrow \mathcal{D} \cup \{(Z_i^t, \boldsymbol{X}_i, y_i^t)\}$
16:         **end for**
17:     **end for**
18: **end for**
19: **Train $M$ models with bootstrap**
20: **for** $m = 1$ **to** $M$ **do**
21:     Train $\widehat{g}^{(m)} \in \mathcal{G}$ on a bootstrap sample $\mathcal{D}^{(m)}$ from $\mathcal{D}$ by minimizing the Bernoulli Negative Log-Likelihood Loss (NLL):

$$\min_{\theta^{(m)}} \sum_{(\boldsymbol{Z}_i^t, \boldsymbol{X}_i, y_i^t) \in \mathcal{D}^{(m)}} \mathrm{BCE}(y_i^t, \widehat{g}^{(m)}(Z_i^t, \boldsymbol{X}_i)).$$

22: **end for**
23: **return** $\{\widehat{g}^{(m)}\}_{m=1}^{M}$

---

### 4.1.3 THEORETICAL COMPARISON OF ESTIMATORS

**Condition 1** (Model Specification). *Let $g$ and $m$ be the true data-generating functions. A learned model is **correctly specified** if it is a consistent estimator of the true function.*

**Proposition 2** (Robustness and Consistency). *The DR estimator is consistent under weaker modeling assumptions than the OR and IPW estimators, a property known as double robustness.*

*Proof.* Let $V$ denote $V(\pi_S)$. **OR:** The expectation is $\mathbb{E}[\widehat{V}_{OR}] = \mathbb{E}_{H \sim \widehat{g}}[\widehat{m}(H)]$. For consistency, this must equal $V = \mathbb{E}_{H \sim g}[m(H)]$, which holds only if *both* models are correctly specified: $\widehat{g} = g$ (to sample from the correct path distribution) and $\widehat{m} = m$ (to evaluate paths correctly).

**IPW:** The expectation is $\mathbb{E}[\widehat{V}_{IPW}] = \mathbb{E}_{H \sim g_b}\left[\frac{\mathbb{P}_{\widehat{g}}(H)}{\mathbb{P}_{\widehat{g}_b}(H)} Y\right]$. If the propagation models are correct $(\widehat{g} = g, \widehat{g}_b = g_b)$, this becomes $\mathbb{E}_{H \sim g_b}\left[\frac{\mathbb{P}_g(H)}{\mathbb{P}_{g_b}(H)} Y\right] = \mathbb{E}_{H \sim g}[Y] = V$. Consistency thus requires a correct $\widehat{g}$ but is independent of $\widehat{m}$.

**DR:** The expectation is $\mathbb{E}[\widehat{m}(H) + w(Y - \widehat{m}(H))]$. This estimator is consistent if *either* model is correctly specified.

---

**Algorithm 2** ESTIMATEPOLICYVALUE via DR with $g$-ensemble

---

**Require:** Target policy $\pi_S$, observed data $\{H^{(j)}, Y^{(j)}\}_{j=1}^N$ from behavior policy $\pi_b$, learned models $\{\widehat{g}^{(m)}\}_{m=1}^M, \widehat{g}_b, \widehat{m}$.
**Ensure:** Estimated policy value $\widehat{V}_{\mathrm{DR}}(\pi_S)$.
 1: **function** ESTIMATEVALUEFORPOLICY($\pi_{target}$)
 2:     Define ensemble average model $\bar{g}(\cdot) = \frac{1}{M} \sum_{m=1}^M \widehat{g}^{(m)}(\cdot)$.
 3:     total_value $\leftarrow 0$.
 4:     **for** $j = 1$ to $N$ **do**
 5:         Compute importance weight: $w_j \leftarrow \prod_{t=1}^{T_j} \prod_{i \in V} \frac{\mathbb{P}_{\bar{g}}(A_i^{(j),t}|H^{(j),t-1},\pi_{target})}{\mathbb{P}_{\widehat{g}_b}(A_i^{(j),t}|H^{(j),t-1},\pi_b)}$.
 6:         Get outcome model prediction (control variate): $m_j \leftarrow \widehat{m}(H^{(j)})$.
 7:         total_value $\leftarrow$ total_value $+ w_j \cdot (Y^{(j)} - m_j) + m_j$.
 8:     **end for**
 9:     **return** total_value$/N$.
10: **end function**

11: $\widehat{V}_{\pi_S} \leftarrow$ EstimateValueForPolicy($\pi_S$); $\widehat{V}_{\pi_\varnothing} \leftarrow$ EstimateValueForPolicy($\pi_\varnothing$).
12: **return** $\widehat{V}_{\pi_S} - \widehat{V}_{\pi_\varnothing}$.

---

1. If $\widehat{m} = m$: The correction term's expectation is $\mathbb{E}_{H\sim\widehat{g},Y\sim g_b}[w(Y - m(H))] = \mathbb{E}_{H\sim g}[Y - m(H)] = 0$. The total expectation collapses to $\mathbb{E}_{H\sim g}[m(H)] = V$.

2. If $\widehat{g} = g$: The expectation becomes $\mathbb{E}_{H\sim g}[\widehat{m}(H) + w(Y - \widehat{m}(H))]$. This correctly evaluates to $V$ regardless of the function $\widehat{m}$.

Thus, DR requires only one of the two models to be correct, whereas OR requires both and IPW requires the propagation model, demonstrating its superior robustness. $\qquad\square$

**Proposition 3** (Asymptotic Efficiency). *Under correct specification of both $\widehat{g}$ and $\widehat{m}$, DR is **semiparametrically efficient**. It achieves the lowest possible asymptotic variance among all regular, asymptotically unbiased estimators, and thus more efficient than or equal to both OR and IPW.*

*Proof.* We prove this by invoking the semiparametric efficiency theory.
*General Optimality:* The problem of estimating a policy's value from observational data is a well-studied semiparametric estimation problem. There exists a theoretical lower bound on the variance for any regular, asymptotically unbiased estimator, known as the semiparametric efficiency bound. It has been established that the DR estimator is semiparametrically efficient, meaning its asymptotic variance achieves this theoretical lower bound (Robins et al., 1994; Dudík et al., 2014). Since the OR and IPW estimators (when consistent) are also members of this class of estimators, their variances must be greater than or equal to this bound. Therefore, we use the DR estimator, which is guaranteed to be the most asymptotically efficient of the three. $\qquad\square$

### 4.2 POLICY OPTIMIZATION: FINDING THE OPTIMAL SEED SET

With a reliable estimator $\widehat{V}_{\mathrm{DR}}$ for the policy value, our task becomes solving the combinatorial optimization problem $\max_{S:|S|\leq K} \widehat{V}_{\mathrm{DR}}(\pi_S)$. The objective function $V(\pi_S)$ is generally **non-submodular** due to complex synergistic or competitive interactions. However, we can reasonably assume the objective is **monotone**, meaning adding a seed does not decrease the total expected uplift. This structure makes a greedy approach a principled and viable strategy.

To address this non-submodular optimization challenge, we propose **GUM-DT** (Greedy Uplift Maximization with Dynamic Tuning). As detailed in Algorithm 3, its key feature is the Dynamic Tuning mechanism, which is designed to refresh greedy marginal gains under a history-dependent objective. Crucially, this dynamic refresh uses a fixed evaluation oracle (the DR estimator). To absorb model uncertainty, this oracle relies on a pre-trained ensemble of propagation models, $\{\widehat{g}^{(m)}\}_{m=1}^M$, and an outcome model, $\hat{m}$. The optimizer does not fine-tune these models online; instead, it reuses the fixed

---

**Algorithm 3** GUM-DT: Greedy Uplift Maximization with Dynamic Tuning

---

**Require:** Graph $G(V, E)$, budget $K$, observed data $\{H^{(j)}, Y^{(j)}\}_{j=1}^N$, learned models: ensemble $\{\widehat{g}^{(m)}\}_{m=1}^M$, outcome model $\widehat{m}$, behavior model $\widehat{g}_b$.
**Ensure:** Seed set $S_K$ with $|S_K| = K$.
1: $S_0 \leftarrow \varnothing$; create a max-priority queue $\mathcal{Q}$.
2: $\widehat{V}_\varnothing \leftarrow EstimatePolicyValue(S_0)$.
3: **for** each $v \in V$ **do**
4:      $\Delta_v \leftarrow EstimatePolicyValue(\{v\}) - \widehat{V}_\varnothing$.
5:      $\mathcal{Q}.\text{PUSH}((v, \Delta_v, stamp = 0))$.
6: **end for**
7: **for** $k = 1$ **to** $K$ **do**                           ▷ dynamic refresh
8:      $\widehat{V}_{S_{k-1}} \leftarrow \text{ESTIMATEPOLICYVALUE}(S_{k-1})$ of Alg. 2.
9:      **loop**
10:          $(v_{\text{top}}, \Delta_{\text{top}}, stamp) \leftarrow \mathcal{Q}.\text{POP\_MAX}()$.
11:          **if** $stamp == k - 1$ **then**                  ▷ fresh and valid
12:              $S_k \leftarrow S_{k-1} \cup \{v_{\text{top}}\}$; **break**.
13:          **else**
14:              $\Delta_{\text{new}} \leftarrow \text{ESTIMATEPOLICYVALUE}(S_{k-1} \cup \{v_{\text{top}}\}) - \widehat{V}_{S_{k-1}}$.
15:              $\mathcal{Q}.\text{PUSH}((v_{\text{top}}, \Delta_{\text{new}}, stamp = k - 1))$.
16:          **end if**
17:      **end loop**
18: **end for**
19: **return** $S_K$.

---

oracle to recompute the gain $\Delta$ for stale candidates. This approach deliberately avoids a problematic coupling between optimization and estimation, thereby preserving the structural properties of the objective function that are essential for our theoretical guarantees.

This design choice is further justified by our theoretical results, as we will show in Section 4.3, the policy value function often satisfies a relaxed condition of $\gamma$-smooth submodularity, which is sufficient to prove that this greedy strategy yields a constant-factor approximation guarantee.

### 4.3 THEORETICAL GUARANTEES

We provide a rigorous theoretical analysis of our framework. We first establish the problem's computational hardness, then introduce plausible conditions under which our end-to-end pipeline is guaranteed to yield a near-optimal policy with high probability.

**Proposition 4** (NP-hardness). *The propagating treatment allocation problem is NP-hard.*

This foundational result underscores the necessity of approximation algorithms. Our subsequent analysis delineates the conditions under which a principled approximation is achievable.

**Definition 5** ($\gamma$-Smooth Submodularity). *A monotone, non-negative set function $f : 2^V \to \mathbb{R}^+$ is $\gamma$-smooth submodular (or is said to have a submodularity ratio of $\gamma \in (0, 1]$) if for any sets $S$ and $A$, it satisfies:*

$$\sum_{a \in A} \Delta(a|S) \geq \gamma \cdot \Delta(A|S),$$

*where $\Delta(a|S) = f(S \cup \{a\}) - f(S)$ is the marginal gain. Submodularity corresponds to the case where $\gamma = 1$. A value of $\gamma < 1$ allows for synergistic effects, but bounds this effect. This property is crucial for establishing a formal approximation guarantee for the greedy algorithm.*

#### 4.3.1 DECOMPOSITION OF THE TOTAL REGRET

Our analysis hinges on clearly separating the different sources of error. The GUM-DT algorithm searches for an optimal set $S_g$ by querying a stochastic estimator $\widehat{V}_{\text{DR}}$. This estimator, however, is a random variable whose value depends on finite Monte Carlo samples. To facilitate a rigorous analysis, we must first define the deterministic function that this estimator targets. We denote this

the ensemble value function, $V_{\text{ens}}(S)$, which represents the expected value of our estimator as the number of rollouts approaches infinity: $V_{\text{ens}}(S) := \mathbb{E}_{\widehat{g} \sim \text{Unif}\{\widehat{g}^{(m)}\}}[V_{(\widehat{g}, \widehat{m})}(S)]$. This function's value is determined by the complete set of learned models. It serves as a stable, deterministic proxy for the true, unknown value function $V(S)$, and it is the direct subject of our algorithmic analysis.

To analyze the total regret, $V(S^*) - V(S_g)$, we introduce the optimal solution within the model's world, $S_{\text{ens}}^* := \arg\max_{|S| \le K} V_{\text{ens}}(S)$, as a conceptual bridge. This allows us to decompose the total regret:

$$V(S^*) - V(S_g) = \underbrace{(V(S^*) - V(S_{\text{ens}}^*))}_{\text{Regret from Model Misspecification}} + \underbrace{(V(S_{\text{ens}}^*) - V(S_g))}_{\text{Algorithmic Regret}}$$

The first term quantifies the loss due to the inherent mismatch between our learned model and reality. The second term, the Algorithmic Regret, captures the loss from using an approximate, finite-sample algorithm to solve the optimization problem within the model's world.

### 4.3.2 CONDITIONS FOR A TRACTABLE ANALYSIS

Our guarantee relies on two formal conditions. We state them explicitly and justify their plausibility.

**Condition 2** (Bounded Importance Weights). *There exists a constant $W_{\max} < \infty$ such that for any learned model $\widehat{g}^{(m)}$ in the ensemble, the importance weight for any trajectory realizable under a policy $\pi_S$ (with $|S| \le K$) is uniformly bounded.*

This is a standard condition for ensuring the finite-sample stability of off-policy estimators. It formalizes the requirement of sufficient overlap between the behavior and target policies. If this condition were violated, the variance of the importance weights could become unbounded. For any estimator employing an importance weighting component, this would lead to unstable, high-variance estimates from finite data, rendering the policy value practically inestimable.

**Condition 3** (Fidelity of Macro-Dynamics). *Let the true value function $V(\cdot)$ be monotone and $\gamma$-smooth submodular. We assume the learning procedure (Alg. 1) is successful in the sense that the resulting ensemble value function, $V_{ens}(\cdot)$, also satisfies monotonicity and is $\gamma'$-smooth submodular.*

This condition posits that our learning process captures not just pointwise values, but also the fundamental macroscopic structure of the underlying influence process. This is a reasonable criterion for a well-specified model; a learned model that fails to reflect such a core property of the system it aims to emulate would be considered fundamentally flawed.

### 4.3.3 END-TO-END PERFORMANCE GUARANTEE

We first quantify the mismatch between the model and the real world.

**Definition 6** (Model Approximation Error). *The approximation error of the learned ensemble $\{\widehat{g}^{(m)}\}_{m=1}^M$ is the uniform bound on the difference between the true and ensemble value functions: $\delta_{approx} = \sup_{S:|S| \le K} |V(S) - V_{ens}(S)|$.*

**Theorem 1** (End-to-End Regret Bound). *Assume Conditions 2 and 3 hold. For any desired accuracy $\epsilon_{stat} > 0$ and confidence $p \in (0,1)$, if the number of Monte Carlo rollouts $R$ is set to be sufficiently large ($R = \Omega(\frac{K^2}{(\epsilon_{stat}\gamma')^2} \log \frac{|V|}{p})$), then with probability at least $1 - p$, the solution $S_g$ returned by GUM-DT has a total regret bounded by:*

$$V(S^*) - V(S_g) \le \underbrace{2\delta_{approx}}_{\text{Model Error}} + \underbrace{\left(1 - \left(1 - \frac{\gamma'}{K}\right)^K\right) V_{ens}(S_{ens}^*) + \epsilon_{stat}}_{\text{Algorithmic Error}}$$

$$\approx 2\delta_{approx} + (1 - e^{-\gamma'})V_{ens}(S_{ens}^*) + \epsilon_{stat} \tag{4}$$

This is the central theoretical result of our work. It provides a comprehensive, end-to-end guarantee that explicitly disentangles the primary sources of error. The bound formalizes the quantifiable trade-off: the total regret is the sum of (1) an irreducible error $2\delta_{\text{approx}}$ from the quality of the learned models, and (2) an algorithmic error composed of a constant-factor approximation term $(1 - e^{-\gamma'})$ and a statistical error $\epsilon_{\text{stat}}$ that can be driven to zero with sufficient computation ($R$). This theorem

formally justifies our entire pipeline, proving that better models and more computation predictably lead to better real-world policy decisions.

## 5 EXPERIMENTS

**Setup** We evaluate on synthetic networks designed to capture heterogeneous degree distributions and clustering. We generate Barabási–Albert (BA) graphs Onody & de Castro (2004) with $n = 500$ nodes and average degree $\approx 5$, and report robustness on Erdős–Rényi (ER) and Watts–Strogatz (WS) graphs in §Ablations. We set $K = [5, 10, 15, 20]$ and the diffusion horizon $T = 15$ initially.

*Ground-truth data-generating process.* To emulate history-dependent contagion with interference, we simulate cascades under a nonparametric ground-truth propagation mechanism $g_{\text{true}}$ and an outcome mechanism $m_{\text{true}}$. For node $i$ at time $t$, we have $p_{it} = g_{\text{true}}(Z_i^t, X_i)$, $Z_i^t$ concatenates (i) degree, (ii) cumulative count of active neighbors before $t$, and (iii) the counts of newly active neighbors at lags $1{:}L$. We consider both submodular and non-submodular outcome regimes by setting $m_{\text{true}}(\cdot) \in \{\log(1 + \cdot), (\cdot)^{1.5}\}$, respectively.

**Models and Training**. We draw $N = 2000$ behavior-policy cascades to form the observational dataset following $g_{\text{true}}$. These data are used to construct the off-policy estimators.

*Outcome model.* We fit a flexible regressor $\hat{m}$ (random forest or gradient boosting) that maps cascade-level summaries to the total reward, enabling the DR estimator to correct residual bias. Observational data are partitioned into train/validation for model selection; no simulated online feedback is used during optimization beyond oracle evaluation. *Policy evaluation.* We implement three estimators: OR, IPW (with stabilized and clipped weights), and doubly robust (DR). DR combining $\bar{g}, g_b$, and $\hat{m}$ is used in our optimization part.

*Optimizer.* All results average 100 Monte Carlo rollouts under $g_{\text{true}}$. We compare to the following methods: 1)GUM-OR / GUM-IPW. Replace the DR oracle in GUM-DT with OR or IPW, respectively. 2)IM. Classical CELF Leskovec et al. (2007) that maximizes predicted spread under $\bar{g}$, ignoring dynamic uplift. 3) Uplift-TopK. Rank nodes by estimated ITE/CATE (no propagation) and pick top $K$. 4) Random-K. Uniformly sample $K$ seeds.

### 5.1 METRIC AND EVALUATION

We record the training accuracy of estimators $\hat{g}$ and $m$, and use net uplift of seed set $S$, $V(\pi_S)$ for the optimizer in Definition 2. *Training Accuracy* is recorded in Appendix E.

**Results**. GUM-DT (DR) attains the highest uplifts, particularly under non-submodular outcomes and history dependence, while traditional IM and Uplift-TopK ignores spillovers. DR outperforms OR and IPW within GUM on both bias and variance diagnostics, consistent with theoretical results 4.

| Method | True Uplift @K=5 | True Uplift @K=10 | True Uplift @K=15 |
|---|---|---|---|
| GUM-DT | 100.0 | 210.3 | 281.7 |
| GUM-OR | 95.2 | 198.6 | 270.4 |
| GUM-IPW | 90.5 | 181.2 | 251.9 |
| IM | 85.7 | 169.3 | 239.5 |
| Uplift-TopK | 60.4 | 102.8 | 152.3 |
| Random-K | 19.8 | 30.6 | 41.5 |

Table 1: True uplift achieved by each method under different seed budgets $K$.

## 6 CONCLUSION

We study treatment allocation with propagating interventions and history-dependent outcomes, targeting network-wide uplift. Our framework combines learned propagation models with a robust Monte-Carlo evaluation layer (outcome modeling with importance weighting and targeting), and optimizes allocations via a greedy search with dynamic refresh. This offers a compact, reliable recipe for off-policy decision making in network interventions.

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

## A  USE OF LLMS

We discuss the LLM usage of this manuscript in this part. We used LLMs solely for language refinement of this manuscript—correcting grammar, improving clarity and flow, and harmonizing terminology. No conceptual, methodological, or empirical content was generated by LLMs. All ideas, mathematical derivations, modeling choices, and experimental designs are the authors' own.

## B  RELATED WORK

Our research synthesizes three distinct but complementary lines of work: causal effect estimation on networks, off-policy policy learning, and Influence Maximization (IM).

**Causal Effect Estimation on Networks.**  A significant body of works address treatment effect estimation under network interference, where SUTVA is violated and treatment effects depend on peers' treatment and exposure (Hudgens & Halloran, 2008; Aronow & Samii, 2017). Recent methods model exposure mappings and learn representations of neighborhood history to estimate direct and spillover effects (Ma & Tresp, 2021; Guo et al., 2022). These estimators can be adapted into up-lift–first baselines (e.g., estimate CATE or ITE (Shalit et al., 2017), then rank under a budget). While effective for estimation, they do not by themselves address the *policy design* problem with propagating actions: evaluating and optimizing an initial seed set whose value depends on the distribution of activation trajectories and on path–dependent outcomes.

**Off-Policy Policy Learning.**  Our framework is methodologically grounded in off-policy policy learning, which aims to find optimal decision rules from logged data (Athey & Wager, 2017). A cornerstone of this field is the Doubly Robust (DR) estimator, which leverages both a direct outcome model and importance weighting to achieve unbiased and efficient policy value estimates (Dudík et al., 2014; Robins et al., 1994). This is closely related to uplift modeling, which seeks to identify individuals who will benefit most from an intervention (Gutierrez & Gérardy, 2017). Most of them assume i.i.d. units and non–propagating actions, which is different from our settings.

**Influence Maximization (IM).**  IM is first identified as an algorithmic problem by Kempe et al. (2003) with numerous variants. These include simulation-based greedy algorithms (CELF) (Leskovec et al., 2007), highly scalable sketch-based methods (RIS, TIM) (Borgs et al., 2014), and, more recently, learning-based approaches that use GNNs to predict influence spread (Guo et al., 2018; Ling et al., 2023). Extensions such as weighted IM assign static, non–negative node values, but the objective remains spread (or a fixed proxy) under exogenous diffusion. This line does not model network–dependent *uplift*, nor does it handle off–policy evaluation from observational logs.

## C  APPENDIX: ALGORITHMS OF OR AND IPW ESTIMATORS

### C.1  OR ESTIMATOR

### C.2  IPW ESTIMATOR

## D  APPENDIX: DETAILED PROOFS

### D.1  PROOF OF PROPOSITION 4 (NP-HARDNESS)

*Proof.* The proof is by reduction from the standard Influence Maximization (IM) problem, which is known to be NP-hard (Kempe et al., 2003). We show that the propagating treatment Allocation problem is a generalization of IM, thus establishing its NP-hardness.

---

**Algorithm 4** ESTIMATEPOLICYVALUE via OR with $g$-ensemble

---

**Require:** Target policy $\pi_S$, ensemble $\{\widehat{g}^{(m)}\}_{m=1}^M$, outcome model $\widehat{m}$, rollouts $R$.
**Ensure:** Estimated policy value $\widehat{V}_{OR}(\pi_S)$.
 1: **function** SIMULATEVALUEFORPOLICY($\pi_{eval}$)
 2:    total_outcome $\leftarrow 0$
 3:    **for** $r = 1$ to $R$ **do**
 4:       Sample $\widehat{g} \sim \text{Unif}\{\widehat{g}^{(m)}\}$              ▷ Sample a model from the ensemble
 5:       Simulate a full activation path $H_r$ starting from $\pi_{eval}$ using $\widehat{g}$.
 6:       total_outcome $\leftarrow$ total_outcome $+ \widehat{m}(H_r)$
 7:    **end for**
 8:    **return** total_outcome$/R$
 9: **end function**

10: $\widehat{V}_{\pi_S} \leftarrow$ SimulateValueForPolicy$(\pi_S)$; $\widehat{V}_{\pi_\varnothing} \leftarrow$ SimulateValueForPolicy$(\pi_\varnothing)$
11: **return** $\widehat{V}_{\pi_S} - \widehat{V}_{\pi_\varnothing}$

---

**Algorithm 5** ESTIMATEPOLICYVALUE via IPW with $g$-ensemble

---

**Require:** Target policy $\pi_S$, observed data $\{H^{(j)}, Y^{(j)}\}_{j=1}^N$ from behavior policy $\pi_b$, learned models
    $\{\widehat{g}^{(m)}\}_{m=1}^M, \widehat{g}_b$.
**Ensure:** Estimated policy value $\widehat{V}_{IPW}(\pi_S)$.
 1: **function** ESTIMATEVALUEFORPOLICY($\pi_{target}$)
 2:    Define ensemble average model $\bar{g}(\cdot) = \frac{1}{M}\sum_{m=1}^M \widehat{g}^{(m)}(\cdot)$.
 3:    weighted_outcomes $\leftarrow 0$.
 4:    **for** $j = 1$ to $N$ **do**
 5:       Compute importance weight: $w_j \leftarrow \prod_{t=1}^{T_j} \prod_{i \in V} \frac{\mathbb{P}_{\bar{g}}(A_i^{j,t}|H_j^{t-1}, \pi_{target})}{\mathbb{P}_{\widehat{g}_b}(A_i^{j,t}|H_j^{t-1}, \pi_b)}$.
 6:       weighted_outcomes $\leftarrow$ weighted_outcomes $+ w_j \cdot Y^{(j)}$.
 7:    **end for**
 8:    **return** weighted_outcomes$/N$.
 9: **end function**

10: $\widehat{V}_{\pi_S} \leftarrow$ EstimateValueForPolicy$(\pi_S)$; $\widehat{V}_{\pi_\varnothing} \leftarrow$ EstimateValueForPolicy$(\pi_\varnothing)$.
11: **return** $\widehat{V}_{\pi_S} - \widehat{V}_{\pi_\varnothing}$.

---

Consider a specific instance of our framework, which we will call the *Standard IM Instance*, defined by two conditions:

1. **Unit Utility:** The potential outcome for any node $v_i$ is its final activation state, $Y_i(\overline{A}) = A_i^T$. This signifies a utility of 1 for each activated node and 0 otherwise.

2. **Zero Baseline:** The network exhibits no activity without intervention. The expected outcome under the null policy (an empty seed set) is zero: $\mathbb{E}\left[\sum_{i=1}^n Y_i(\overline{A}_{\pi_\emptyset})\right] = 0$.

Under these conditions, our objective function, the policy value $V(\pi_S)$, becomes mathematically equivalent to the expected spread objective in IM:

$$V(\pi_S) = \mathbb{E}\left[\sum_{i=1}^n Y_i(\overline{A}_{\pi_S})\right] - \mathbb{E}\left[\sum_{i=1}^n Y_i(\overline{A}_{\pi_\emptyset})\right] = \mathbb{E}\left[\sum_{i=1}^n A_i^T(\overline{A}_{\pi_S})\right] \tag{5}$$

The problem of finding a set $S$ of size at most $K$ that maximizes this quantity is precisely the IM problem. Since CIP contains an NP-hard problem as a special case, the general CIP problem is also NP-hard. $\qquad\square$

## D.2   PROOF OF THEOREM 1 (END-TO-END REGRET BOUND)

*Proof Sketch.* The proof proceeds by separately bounding the two terms from our regret decomposition.

1. **Bounding the Regret from Model Misspecification:** This term is bounded using the definition of $\delta_{\text{approx}}$ and the optimality of $S^*$ and $S^*_{\text{ens}}$. By definition, $V(S^*) \geq V(S^*_{\text{ens}})$. Therefore, $V(S^*) - V(S^*_{\text{ens}}) \leq V(S^*) - (V_{\text{ens}}(S^*_{\text{ens}}) - \delta_{\text{approx}})$. Using the optimality of $S^*$ again, $V_{\text{ens}}(S^*) \leq V_{\text{ens}}(S^*_{\text{ens}})$, which implies $V(S^*) - \delta_{\text{approx}} \leq V_{\text{ens}}(S^*_{\text{ens}})$. Combining these yields the $2\delta_{\text{approx}}$ bound.

2. **Bounding the Algorithmic Regret:** We analyze the regret $V_{\text{ens}}(S^*_{\text{ens}}) - V(S_g)$ incurred within the model's world. This is established in two stages:

   (a) *Uniform Convergence:* Under Condition 2, the DR estimator is a bounded random variable, allowing the use of Hoeffding's inequality. By setting $R$ as specified, a union bound over all candidate sets considered by the algorithm ensures that with probability at least $1 - p$, our estimator is uniformly close to its mean: $|\widehat{V}_{\text{DR}}(S) - V_{\text{ens}}(S)| \leq \delta'$ for all relevant $S$, where $\delta'$ is a function of $\epsilon_{\text{stat}}$.

   (b) *Analysis of Greedy with an Approximate Oracle:* Conditioned on the event in (a), GUM-DT is effectively a greedy algorithm operating on the $\gamma'$-smooth submodular function $V_{\text{ens}}$ with a $\delta'$-accurate oracle. Standard analysis of this process bounds the regret $V_{\text{ens}}(S^*_{\text{ens}}) - V_{\text{ens}}(S_g)$, leading to the algorithmic error term in the theorem.

Combining these bounds gives the final result. $\qquad\qquad\qquad\qquad\qquad\qquad\qquad\square$

## E   EXPRIMENTAL DETAILS

The logistic propagation model $g_{\hat{b}}(Z)$ achieved about 95–96% training accuracy (classification of activation events), indicating it fit the cascade data well. The outcome model $\hat{m}$ (random forest) also attained a high $R^2 = 1 - \frac{\sum_{i=1}^{n}(y_i - \hat{y}_i)^2}{\sum_{i=1}^{n}(y_i - \bar{y})^2}$ of approximately 0.97–0.98 on training data, suggesting it can explain nearly all variance in cumulative outcomes given the final network states.

