# OpenReview forum: "Learning to Allocate Propagating Treatments: Maximize Uplift under Network Interference"
_ICLR.cc/2026/Conference — ICLR 2026 Conference Withdrawn Submission_

### Official Review · Reviewer_hJ9z · 2025-10-31

**Soundness:** 2
**Presentation:** 1
**Contribution:** 2
**Rating:** 2
**Confidence:** 4

**Summary:**

This paper introduces a new method for optimizing treatment allocation when treatment effects can propagate through a network. In addition, the authors theoretically show the regret bound for their method.

**Strengths:**

The authors investigate an interesting and relevant problem: optimizing treatment allocation when treatment effects can propragate through the network.

**Weaknesses:**

The authors seem to ignore some related works in areas that are highly relevant to their work.

- I am not convinced that the problem they are investigating is different from Influence Maximization (IM), as the authors claim in the Introduction (Line 45). As I understand it, the goal in IM is to select a set of entities so as to maximize the total influence spread. While these papers often make assumptions regarding how influence spreads through the network that are different from the assumptions in this paper, I believe the core problem is the same. Additionally, there are a lot more relevant papers in IM that try to estimate influence spread or diffusion models using data. For example, [1-4].
- There are a lot more relevant works related to Causal effect estimation in networks, such as [5-8]. Additionally, [9] even proposes a method for estimating treatment effects in networks over time, and [10] does something very similar to your work.

Overall, the authors are very vague about the motivation for their modelling choices and how they implemented everything. Therefore, it is very hard to judge whether their method makes sense. Below, I list some issues.

- I do not fully understand why you model $m$ directly as the sum of outcomes. Is there a reason you do not predict the outcomes of each entity separately and then sum them?

- To encode the history, a time-channel GNN is used. However, there are no references or explanations about what this is and why this is the best choice. Related to this point, the authors use GNN to model $m$. I find it hard to understand how this works exactly. What are the inputs, operations, and outputs of this GNN?

- The authors never describe how they implement the baselines used in the experiment. For example, what diffusion model is used vor CELF?

- The data-generating process is only introduced on a very high level, making it impossible to understand how exactly it works.



[1] Goyal, A., Bonchi, F., Lakshmanan, L.V., 2010. Learning influence probabilities in social networks, in: Proceedings of the third ACM International conference on Web search and data mining, pp. 241–250.

[2] Saito, K., Nakano, R., Kimura, M., 2008. Prediction of information diffusion probabilities for independent cascade model, in: International conference on knowledge-based and intelligent information and engineering systems, Springer. pp. 67–75.

[3] Aral, S., Dhillon, P.S., 2018. Social influence maximization under empirical influence models. Nature human behaviour 2, 375–382.
[4] Bourigault, S., Lamprier, S., Gallinari, P., 2016. Representation learning for information diffusion through social networks: an embedded cascade model, in: Proceedings of the Ninth ACM international conference on Web Search and Data Mining, pp. 573–582.

[5] Laura Forastiere, Edoardo M Airoldi, and Fabrizia Mealli. Identification and estimation of treatment and interference effects in observational studies on networks. Journal of the American Statistical Association, 116(534):901–918, 2021.

[6] Song Jiang and Yizhou Sun. Estimating causal effects on networked observational data via representation learning. In Proceedings of the 31st ACM International Conference on Information 2022.

[7] Elizabeth L Ogburn, Oleg Sofrygin, Ivan Diaz, and Mark J Van der Laan. Causal inference for social network data. Journal of the American Statistical Association, 119(545):597–611, 2024.

[8] Weilin Chen, Ruichu Cai, Zeqin Yang, Jie Qiao, Yuguang Yan, Zijian Li, and Zhifeng Hao. Doubly robust causal effect estimation under networked interference via targeted learning. In International Conference on Machine Learning, 2024.

[9] Song Jiang, Zijie Huang, Xiao Luo, and Yizhou Sun. Cf-gode: Continuous-time causal inference for multi-agent dynamical systems. In Proceedings of the 29th ACM SIGKDD Conference on Knowledge Discovery and Data Mining, pp. 997–1009, 2023.

[10] Daan Caljon, Jente Van Belle, Jeroen Berrevoets, and Wouter Verbeke. Optimizing treatment allocation in the presence of interference. European Journal of Operational Research, 2025.

**Questions:**

I invite the authors to interpret the Weaknesses section as questions.

---

### Official Review · Reviewer_e2gV · 2025-11-01

**Soundness:** 1
**Presentation:** 1
**Contribution:** 2
**Rating:** 0
**Confidence:** 4

**Summary:**

The papers propose a double-robust estimator and a dynamic turning algorithm to estimate the causal policy outcome of a dynamic network model in an off-policy fashion. The authors formalize the causal setup for such a system whose potential outcome has both network interference and temporal dependency. The authors also provided some theoretical analysis on the performance of the proposed method.

**Strengths:**

The problem setup is interesting with both network inference and temporal dependence. The paper has its originality in putting it under a causal framework to assess its causal properties, including the identifiability of the causal estimands.

**Weaknesses:**

The major weakness of this paper:
1. The paper is hard to read and quite unclear. Although the authors list all assumptions and conditions, the paper is a stacking of definitions/assumptions/conditions with minimal explanations and elaboration on the subject.
2. The paper's contribution to causal inference is marginal. I appreciate that the authors introduced the causal framework and discussed important causal conditions, including Rubin's potential outcome framework and ignorability assumption. But the paper is essentially working on an off-policy evaluation problem for temporal network data.
3. Lots of assumptions/conditions/theorems lack justification. Some proofs are "proof sketches" and existing proofs are far from rigorous.

**Questions:**

1. Assumption 2 implies that every node has a positive self-activation (i.e. it is activated when all its neighbors are not) probability? Is that expected?
2. In Condition 1, I'm confused with the expression "is a consistent estimator to some function". Is it pointwise consistent, uniformly consistent or consistent under a functional norm？And why introduce the notations $g$ and $m$ here?
3. In Proposition 2, does the "consistency" here means "unbiasedness"? In addition, a pointwise consistency of $\hat g$ to $g$ does not guarantee unbiasedness nor L1 convergence. The latter requires a stronger condition like uniform integrability.
4. The second part of condition 3 is a strong assumption. Any justification on why it is plausible for it to be true?
5. Proof of Theorem 1 is incomplete. Necessary elaboration is missing.

---

### Official Review · Reviewer_Gr9p · 2025-11-05

**Soundness:** 3
**Presentation:** 2
**Contribution:** 2
**Rating:** 6
**Confidence:** 3

**Summary:**

This paper investigates the problem of causal uplift maximization in networks with propagating treatments, where interventions on one unit may influence others via network connections. To enable reliable off-policy evaluation in this setting, the authors develop a tailored doubly robust (DR) estimator that incorporates ensembling and targeted techniques to improve stability and performance under interference. Building on this estimator, they introduce a dynamic-greedy allocator that iteratively updates marginal treatment gains using the targeted DR oracle. The authors further provide theoretical guarantees for this allocation strategy

**Strengths:**

1. The paper presents a novel framework that effectively integrates causal inference with a dynamic-greedy allocation strategy.
2. This paper is very well-organized, easy to follow, and clearly presents its contributions.

**Weaknesses:**

1. While the paper includes empirical comparisons with classical baselines using simulations on synthetic networks, additional experiments—particularly on real-world network data—are needed to more convincingly demonstrate the practical effectiveness and robustness of the proposed approach.

**Questions:**

1. Could the authors provide additional intuition or illustrative examples to clarify the usage of Path-dependent Potential Outcomes? A more intuitive explanation would help readers better understand how treatment propagation and historical dependencies affect outcome definitions in this context.
2. Does the consistency of the proposed framework depend on correct specification of both the propagation model and the outcome model?

---

### Official Review · Reviewer_CG1F · 2025-11-06

**Soundness:** 3
**Presentation:** 2
**Contribution:** 3
**Rating:** 4
**Confidence:** 2

**Summary:**

This paper studies treatment allocation on networks where interventions propagate and outcomes depend on neighbors’ states, causing network interference and history-dependent dynamics. Unlike standard uplift modeling or influence maximization, the goal here is to select a seed set that maximizes net uplift in outcomes relative to a baseline policy, considering multi-step diffusion and spillover effects. The major technique the authors uses are causal inference and combinatorial optimization. Although the idea is very innovative, I can not accept the paper because of several issues mentioned in the weakness and questions sections. I will reconsider my decision if my concerns can be appropriately handled.

**Strengths:**

1. Theoretical Guarantees. The regret decomposition is elegant and clarifies where errors come from (model vs. algorithm).

2. Empirical Performance. Experimental results convincingly show advantage over strong baselines.

3. The major technique the authors used in these paper are causal inference and combinatorial optimization, which is pretty novel.

**Weaknesses:**

1. Synthetic Data Only. Experiments are performed only on synthetic cascades. Lack of real-world evaluation weakens external validity.

2. Clarity of Algorithmic Complexity. The computational cost of each marginal evaluation is not quantified; complexity could be large because every candidate add requires Monte Carlo estimation.

3. Poor Presentation. The sequence of algorithms and proof are somehow entangled together, you may want to separate section 4 into 2 sections so that you can first finish showing the methodology and then focus on theoretical analysis.

4. Difficult to understand for general readers. It is great for the authors to combine causal inference and combinatorial optimization technique to solve the problem, but it seems the article is very difficult to follow. Many notations are not clearly explained with assumptions which may not be practical. The authors should give some simple examples right after formulating the problem and clearly explain in these specific examples, what these notations mean and why the assumptions hold. This will convince the readers and make them much easier to understand the problem.

**Questions:**

1. Are there any real-world application? I suggest the author to give some simple examples after formulating the problem. This will give the readers a clear understanding of the problem.

2. Is it possible to give the computationally complexity of each algorithm? The propagating treatment allocation problem is NP-hard, then what's the gain if we use approximation algorithm? Also, It is a 2-approximation or what? It seems Theorem 1 is trying to answer these questions, but a clear explanation may be needed after presenting the theorem.

3. I suggest the author to separate section 4 into 2 sections so that both the methodology and the theoretical analysis can be more clear.

4. The idea of combining causal inference and combinatorial optimization technique to solve the problem is very interesting, but the authors need to think about a better way to help the reader easily follow the presentation.

5. Small typo. Line 109 "depend on on".

---

### Note · Authors · 2025-11-15

I have read and agree with the venue's withdrawal policy on behalf of myself and my co-authors.